# RealMath: A Continuous Benchmark for Evaluating Language Models on Research-Level Mathematics

**Jie Zhang**    **Cezara Petrui**    **Kristina Nikolić**    **Florian Tramèr**

ETH Zurich

## Abstract

Existing benchmarks for evaluating mathematical reasoning in large language models (LLMs) rely primarily on competition problems, formal proofs, or artificially challenging questions—failing to capture the nature of mathematics encountered in actual research environments. We introduce REALMATH, a novel benchmark derived directly from research papers and mathematical forums that assesses LLMs' abilities on authentic mathematical tasks. Our approach addresses three critical challenges: sourcing diverse research-level content, enabling reliable automated evaluation through verifiable statements, and designing a continually refreshable dataset to mitigate contamination risks. Experimental results across multiple LLMs reveal surprising capabilities in handling research mathematics compared to competition problems, suggesting current models may already serve as valuable assistants for working mathematicians despite limitations on highly challenging problems. The code and dataset for REALMATH are publicly available[1].

## 1 Introduction

The mathematical capabilities of Large Language Models (LLMs) have become a critical lens for assessing their reasoning and knowledge retention abilities. Although considerable effort has been invested in evaluating LLMs in basic mathematics [5, 11], competition-level mathematics [8, 10, 17, 19, 20] and formal proof generation [23, 24, 25], these evaluations may not adequately reflect the potential utility of LLMs in real-world mathematical research contexts.

Current mathematical benchmarks fall predominantly into three categories: (1) those derived from course materials [5, 11] or competitive examinations [8] (e.g., IMO or AIME) that offer abundant problems with solutions; (2) those centered on formal theorem proving where verification can be automated [24, 25]; (3) those designed by mathematical experts to be as challenging as possible [9, 16]. However, these benchmarks capture only a narrow slice of mathematical practice. The mathematics encountered "in the wild"—particularly in research settings—differs substantially from competition problems in structure and topics, rarely relies on formal proofs, and considers statements and results that are not (exclusively) designed to be maximally challenging.

This disconnect raises a fundamental question:

> *How effective might LLMs be as assistants for practicing mathematicians today?*

To address this question, we introduce a novel benchmark designed to evaluate LLMs on research-level mathematics extracted directly from the literature. Constructing such a benchmark presents three significant challenges:

---

[1]Code available at: Github; Dataset available at: Huggingface.

39th Conference on Neural Information Processing Systems (NeurIPS 2025) Track on Datasets and Benchmarks.

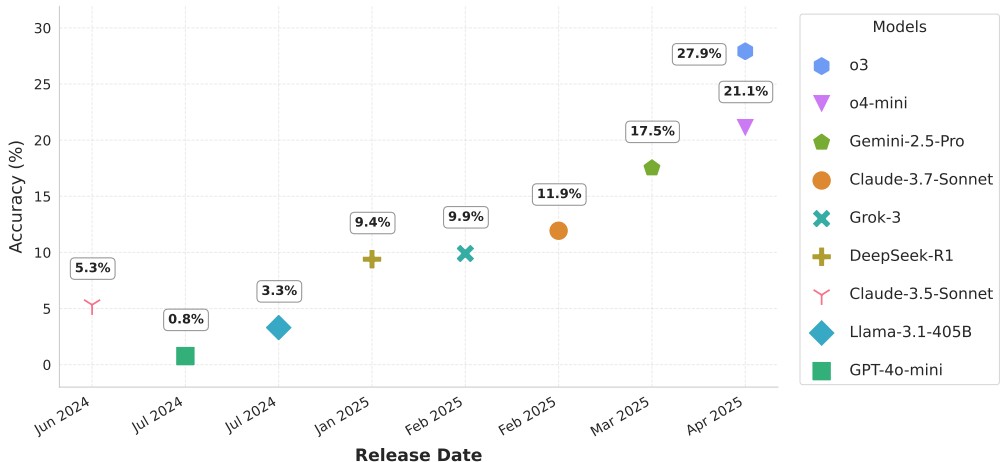

Figure 1: LLM performance on a hard subset of REALMATH from arXiv Mathematics papers.

First, we need to *source authentic content* that faithfully represents the diversity and complexity of contemporary mathematical research.

Second, we require *a reliable method for assessing correctness*. Unlike competition problems with standardized solutions or problems with formal proofs, research mathematics presents evaluation difficulties: human expert validation is resource-intensive and limits scalability, while using LLMs as judges introduces reliability concerns.

Third, we must address *test set contamination over time*, as mathematical content incorporated in benchmarks might be absorbed into training datasets for future models.

In this paper, we present REALMATH, a math benchmark that addresses these challenges through:

1. A data pipeline that extracts verifiable mathematical statements from research papers (e.g., arXiv) and mathematical forums (e.g., Stack Exchange), creating a rich corpus of research-level content.

2. An evaluation methodology focused on verifiable answers rather than proof assessment, allowing for automated correctness checking.

3. A continually refreshable dataset design that leverages the vast and growing body of mathematical literature, allowing for regular updates with new content to mitigate contamination concerns.

We systematically transform mathematical statements into question-answer pairs, preserving surrounding context necessary for comprehension. For example, a theorem stating that "*For $n \geq 1$, the number of shallow, 123-avoiding centrosymmetric permutations of length $n$ is $\frac{n^2}{4} + 1$ when $n$ is even*" [1] becomes a question, "*What is the number of 123-avoiding centrosymmetric permutations of length $n$ when $n$ is even?*", with a verifiable answer, "*$\frac{n^2}{4} + 1$*", with relevant definitions and notation from the source material given as context.

From approximately 9,000 mathematics-related academic papers collected over nine months, our automated pipeline curated 633 high-quality samples and can generate over 70 new samples each month. Our evaluation of frontier LLMs reveals interesting patterns in mathematical capabilities that differ from those observed in other recent benchmarks. In particular, models demonstrate stronger performance on our research-mathematics benchmark (see Figure 1) than on deliberately challenging datasets such as math competitions or FrontierMath [9], suggesting that LLMs may already provide valuable assistance in research contexts, even though they cannot solve the most advanced problems.

Although our approach focuses on statement verification rather than proof generation or verification, it nonetheless provides a valuable signal about LLMs' potential as mathematical assistants. Our findings suggest that current models may already serve as useful tools in mathematical research contexts, even as they continue to struggle with the most challenging mathematical problems.

To summarize, this work makes several contributions: it establishes a new paradigm for evaluating the mathematical capabilities of LLMs using organic research Mathematics; it provides a sustainable methodology for benchmark creation that resists contamination; and it offers insights into the relative strengths of current models on tasks representative of real mathematical practice.

## 2 Related Work

**Math benchmarks sourced from exams and competitions.** Recent years have seen rapid progress in evaluating LLMs on mathematical tasks. Early data sets were sourced from school materials or entry-level math competitions such as GSM8K [5], GHOSTS [7], or MATH [11], and are approaching saturation. In response, new benchmarks now source questions from advanced math competitions such as the IMO [2, 15, 19], AIME, or the Putnam competition [20].

**Math benchmarks sourced from research experts.** A growing line of work seeks to evaluate mathematical reasoning on research-level questions. FrontierMath [9] (and part of HLE [16]) evaluate LLMs on extremely challenging problems crafted by expert mathematicians. These benchmarks focus on a narrow slice of current mathematical practice (at the frontier of human expertise), and are extremely labor-intensive to curate. To guard against test set contamination, FrontierMath keeps the test set private, which introduces barriers to reproducibility. In contrast, REALMATH benchmark consists of questions spanning the full range of mathematical research practice, and can be automatically refreshed as new research is published.

**Benchmarks for theorem proving.** Orthogonal to our work are benchmarks focused on formal proof generation and machine-verifiable mathematics, such as LeanDojo [24] and MiniF2F [25]. These evaluate LLMs' ability to produce formal proofs within interactive theorem provers, supporting automated verification of correctness. While such work is critical for advancing formal methods and proof automation, it is not representative of most mathematical research that is not fully formalized.

**Benchmarks for LLM capabilities "in the wild".** Our work complements a growing body of research that aims to evaluate the capabilities of LLMs on real-world tasks rather than well-curated proxies. For example, instead of evaluating LLMs' coding abilities on programming competitions [12], projects such as SWE-bench [13], Lancer [14], or BaxBench [21] focus on assessing LLMs in real-world software engineering tasks. A similar shift can be observed in the evaluation of cyber-offensive capabilities of LLMs, where a number of recent benchmarks focus on evaluating real exploit capabilities [4, 6] rather than the ability to solve curated capture-the-flag competitions [18].

## 3 Building a Research-Level Mathematical Benchmark

### 3.1 Design Criteria

To ensure high quality of our collected data, we establish the following design desiderata:

1. **Real-world application focus**: Our collection methodology prioritizes mathematical problems that genuinely represent those encountered in practical scenarios. We source problems from academic research publications (e.g., arXiv preprints), where researchers tackle mathematical challenges to advance scientific knowledge, and from educational platforms (e.g., Mathematics Stack Exchange), where learners engage with mathematical concepts to develop their understanding and problem-solving capabilities. Unlike competition-oriented benchmarks, we focus on representative mathematical tasks encountered in actual practice, emphasizing procedural and technique-driven aspects rather than contest-style ingenuity.

2. **Automated verification**: We generate *constructive* problems (as in [2]) with clear, unambiguous verification criteria, typically problems with a single, exact numerical or symbolic answer. This approach excludes problems that admit multiple solutions or involve qualitative assessments such as inequalities (e.g., lower bounds, upper bounds) or asymptotic relations, which would complicate the verification process and potentially introduce ambiguity. This choice also means that we omit statements where the main difficulty lies in finding a (non-constructive) proof (e.g., if a paper has a theorem that says "P $\neq$ NP", then the interesting part is solely the proof and not the construction of the statement itself).

3. **Continuous acquisition**: Leading LLMs may be trained on internet data, so it is important to avoid data contamination. To ensure this, we collect data exclusively from publicly available sources such as arXiv and StackExchange, where automatic collection is feasible with minimal human intervention. After each new model release, we can automatically gather fresh data from the internet to evaluate the model, ensuring that the evaluation set remains uncontaminated and up-to-date.

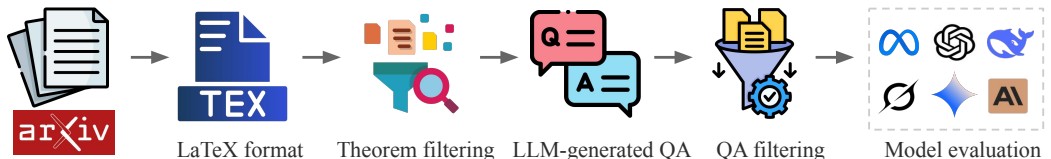

LaTeX format    Theorem filtering    LLM-generated QA    QA filtering    Model evaluation

Figure 2: The data collection pipeline for arXiv papers. The core step is to ensure that each extracted theorem from arXiv papers has a single, exact answer. To maintain data quality, we apply filtering mechanisms, e.g., prompting an LLM to discard trivial samples that can be easily solved.

## 3.2 An Automatic and Refreshable Data Collection Pipeline

We present a detailed walkthrough of our data collection pipeline in Figure 2. Throughout the remainder of this paper, we primarily use data from arXiv papers to illustrate our methodology. A more detailed discussion of the Stack Exchange dataset is provided in Appendix A.5. As a concrete example, we use a subset of 4,000 arXiv papers selected from May to September 2022 to illustrate each stage of the data collection pipeline.

| Step | # Usable Samples |
|---|---|
| Retrieve papers | 05/2022 – 09/2022 (4,000 papers) |
| Extract LaTeX source | 3,922 papers |
| Extract theorems | 14,747 theorems |
| Identify constructive theorems with fixed answers | 407 theorems |
| Generate question-answer (QA) pairs | 401 QA pairs |
| Filter trivial questions | 280 QA pairs |

Table 1: An example of the pipeline steps and the number of usable samples at each stage.

Our pipeline consists of five stages:

1. **Retrieve papers.** We collect all mathematics-related arXiv papers within a specified time window. This process involves querying the arXiv API for papers in categories such as Mathematics or Computer Science. To illustrate, we can acquire roughly 4,000 papers from Mathematics over 5 months, from May to September 2022.

2. **Extract LaTeX source.** For each paper, we download and parse the original LaTeX source files to accurately preserve the mathematical notation. Due to occasional errors in the arXiv API responses or issues with file availability (e.g., missing or corrupted LaTeX sources), the number of successfully processed papers is often lower than the total initially retrieved. In our case, out of 4,000 papers, we obtained 3,922 usable samples.

3. **Identifying constructive theorems with fixed answers.** We use an LLM[2] to identify theorems that involve the construction of a single exact answer. The LLM analyzes each extracted theorem and classifies it based on whether it presents a clear mathematical statement with an unambiguous result. We specifically exclude theorems that involve inequalities or those that admit multiple solutions, e.g., theorems of the type *"If condition A is satisfied, then relation B happens"* but where *condition A* is not the unique one for *B* to happen, and *relation B* is not an equality or fixed-answer numerical relation. From the 3,922 processed papers, we extracted 407 theorems that were classified as high-quality by our judge model, out of 14,747 theorems detected in total.

4. **Generate question-answer (QA) pairs.** For each selected theorem, we use an LLM to convert it into a question–answer pair. LLMs will always attempt to generate a QA pair, even if the generated question–answer is easy to answer. For 407 theorems, we obtained 401 QA pairs.

5. **Filtering trivial questions.** We also implement a post-processing stage where an LLM reviews each generated QA pair. This step filters out low-quality samples, e.g., those with easily guessable answers, or where the answer is obvious from the context fed to the LLM (it is common, for instance, for the paper's introduction to state a theorem's result informally). Finally, we obtained 280 QA pairs that are ready for use in the evaluation phase.

---

[2]By default, we use OpenAI's o3-mini as the judge model. Detailed prompts are provided in Appendix A.1.

For each test sample, we provide the ***context*** (i.e., all relevant text preceding the theorem) along with an LLM-generated ***question*** to frontier LLMs. A response is considered correct only if it exactly matches the ground truth ***answer***. We present some detailed examples in Figure 3.

---

**Theorem to QA Conversion**

| **Original Theorem** | **Generated QA Pair** |
|---|---|
| *(Theorem 1.1 in [3])* Let $q = p^{2t}$, where $p \equiv 3 \pmod 4$ is a prime and $t$ is a positive integer. Then, the number of cliques of order 3 in the Peisert graph $P^*(q)$ is given by $k_3(P^*(q)) = \frac{q(q-1)(q-5)}{48}$. | **Q:** Let $q = p^{2t}$, where $p \equiv 3 \pmod 4$. What is the number of cliques of order 3 in the Peisert graph $P^*(q)$, expressed as a function of $q$? 

 **A:** $\frac{q(q-1)(q-5)}{48}$ |
| *(Theorem 22 in [22])* The automorphism $\varphi_\alpha \circ (\Phi, \wedge d(\Phi))$, where $\alpha \in \mathbb{C}^*$, can be lifted to the $\Pi$-symmetric super-Grassmannian $\Pi\mathrm{Gr}_{2k,k}$, where $k \geq 2$, if and only if $\alpha = \pm i$. | **Q:** Determine the value(s) of $\alpha \in \mathbb{C}^*$ for which the automorphism $\varphi_\alpha \circ (\Phi, \wedge d(\Phi))$ can be lifted to the super-Grassmannian $\Pi\mathrm{Gr}_{2k,k}$ for $k \geq 2$. 
 **A:** $\alpha = \pm i$ |

---

**Filtered Out Examples**

**Example 1:** Let $\mathbb{X}$ be a uniformly convex Banach space. Then $\Gamma(\mathbb{X}) < \frac{1}{2}$.

➤ *LLMs may respond with weaker but still valid bounds (e.g., $\Gamma(\mathbb{X}) < 1$), which are not equivalent. Stronger answers (e.g., $< 0.4$) require more advanced verification from the judge model.*

---

**Example 2:** For any convex body $K$ in $\mathbb{R}^n$, **there exists** $\Phi \in \mathrm{GL}(n)$ such that:

$$\frac{S(\Phi K)^n}{|\Phi K|^{n-1}} \leq \frac{S(\Delta^n)^n}{|\Delta^n|^{n-1}} = \frac{n^{3n/2}(n+1)^{(n+1)/2}}{n!}.$$

➤ *This is a non-constructive existence theorem. It cannot be converted into a fixed-answer QA that could easily be verified by an LLM judge.*

Figure 3: Illustration of the theorem-to-QA conversion process and samples that are filtered out. The top panel shows examples of high-quality question-answer pairs generated from mathematical theorems that contain fixed, verifiable answers. The bottom panel provides examples of theorems that were filtered out due to ambiguity or the lack of a fixed answer.

This pipeline ensures that our benchmark remains current, uncontaminated, and representative of real mathematical challenges faced in research. By automating the collection process, we can continuously refresh the dataset with new mathematical problems as they emerge in the research community.

**Discussion on label noise.** The two primary sources of noise in our data pipeline are: (1) the inherent quality of the source materials; and (2) the reliability of using LLMs to evaluate and transform mathematical content. Samples from Mathematics Stack Exchange, in particular, tend to be of lower quality: Questions are often poorly formulated, contain mathematical inaccuracies, or lack answers on the forum. As a result, obtaining high-quality samples from this source requires filtering through a much larger volume of user-submitted content. Additionally, our pipeline relies on LLMs to assess the quality of extracted theorems and generate QA pairs. Thus, any limitations or biases in the LLM's evaluation capabilities may introduce further inaccuracies into the dataset.

Overall, our pipeline yields more than 94% high-quality samples from arXiv papers, without relying on any human annotation. For example, in the Mathematics category, we processed more than 9,000

Table 3: Frontier LLMs achieve strong performance at answering mathematical questions extracted from research papers and math forums. See Figure 4 and Appendix B.2 for a more fine-grained evaluation based on the estimated difficulty of questions.

| Dataset | o3 | o4 mini | Gemini 2.5-pro | Deepseek R1 | Grok 3 | Claude 3.7-Sonnet | Claude 3.5-Sonnet | Llama 3.1-405B | GPT 4o-mini |
|---|---|---|---|---|---|---|---|---|---|
| `Math.arXiv` | **49.1** | 43.4 | 32.5 | 30.5 | 29.5 | 34.1 | 18.3 | 16.4 | 12.5 |
| `CS.arXiv` | **44.1** | 42.3 | 25.2 | 31.5 | 25.2 | 31.5 | 16.2 | 15.3 | 7.2 |
| `Math.StackExchange` | 70.7 | **70.8** | 60.9 | 62.2 | 54.8 | 61.1 | 37.6 | 32.1 | 40.8 |

papers and extracted 633 theorems that were classified as high-quality by our judge model. We then manually reviewed each sample and found that approximately 6% did not meet our quality criteria; these were filtered out. These results suggest that our pipeline can reliably produce high-quality data with minimal human intervention, even when working with imperfect real-world sources (see Appendix A.3 for more details).

Although our data construction inherently assumes that the mathematical theorems sourced from arXiv papers are *correct*, this is, of course, not guaranteed. ArXiv papers have not necessarily been peer-reviewed, and errors or ambiguous claims may be present.[3] On the one hand, incorrect statements could produce some noise in our results (which we hypothesize to be small). On the other hand, we view possible ambiguities in statements or notation as beneficial, since it is truly representative of mathematics "in the wild", compared to the more polished and vetted content found in formal competition settings.

## 4 Experiments

### 4.1 Experimental Setup

We use the data pipeline described in Section 3 (and Appendix A) to extract over 1,200 QA pairs from three data sources, `Math.arXiv`, `CS.arXiv`, and `Math.StackExchange` (see Table 3 for summary statistics). We then evaluated multiple frontier models—OpenAI o3 and o4-mini, Claude 3.7 Sonnet, Gemini 2.5 Pro, Grok-3, and DeepSeek-R1—across these datasets. For comparison, we also include several earlier models, such as GPT-4o mini, Claude 3.5 Sonnet, and LLaMA-3.1-405B.

Table 2: Summary of final datasets used for evaluation.

| Dataset | Time Span | # QA pairs |
|---|---|---|
| `Math.arXiv` | 05/2022 – 09/2022 12/2024 – 03/2025 | 633 |
| `CS.arXiv` | 05/2022 – 10/2023 | 111 |
| `Math.StackExchange` | 04/2024 – 03/2025 | 542 |

The input format varies slightly across datasets to match their structure. For `CS.arXiv` and `Math.arXiv`, each model receives the LLM-generated question along with the full relevant context preceding the target theorem, typically spanning from the introduction up to (but excluding) the theorem itself. In contrast, for `Math.StackExchange`, the input consists solely of the LLM-generated question, reflecting the format of the original forum posts.

### 4.2 Main Experimental Results

**Results on frontier LLMs.** Table 3 highlights the performance of the wide range of LLMs that we tested on our three datasets. The ten models under analysis show variation in accuracy, with o3 and o4-mini leading on all datasets. More specifically, o3 achieved the highest accuracy on both `CS.arXiv` and `Math.arXiv`, whereas o4-mini slightly outperformed it on the `Math.StackExchange` samples. Other models, such as Deepseek-R1 and Gemini-2.5-pro, showed mid-tier performance, with accuracy percentages ranging between 25.6% – 32.5% on the arXiv datasets, compared to higher accuracy levels of 60.9% – 62.2% on `Math.StackExchange`.

---

[3]An alternative would be to limit the pipeline to peer-reviewed sources, but this would significantly reduce the diversity and coverage of available material.

We acknowledge that our full benchmark comes with relatively high initial performance, which may raise questions about its durability. In fact, a common trend in recent benchmarks has been to design problems that are as challenging as possible, resulting in near-zero performance [9, 16]. However, since our benchmark aims to track performance on *real, organic* mathematical research tasks, we do not view this as an issue. While some benchmarks track LLMs' nascent ability to solve (a few) of the most challenging math problems, performance increases on REALMATH may be more indicative of the utility of LLMs on common mathematical research tasks. Nevertheless, in the following experiment, we also show that the difficulty of questions in our datasets is highly nonhomogeneous and that there is a "hard" subset on which current LLMs perform much worse.

**Breakdown by difficulty levels.** We analyze model performance across questions of varying difficulty levels. To determine these difficulty levels, we evaluated how older, weaker models performed on the dataset samples. Based on their performance, we categorized the samples into different difficulty levels, as detailed in Appendix B.2. As shown in Figure 4, accuracy consistently declines as question difficulty increases. For example, the top-performing model, o3, achieves 97.5% accuracy on easy questions, dropping to 81.4% on medium questions and further to just 27.9% on hard questions. A similar downward trend is observed for Gemini-2.5-Pro and DeepSeek-R1. This pattern highlights the significant challenge current models face in handling more complex and research-level mathematics.

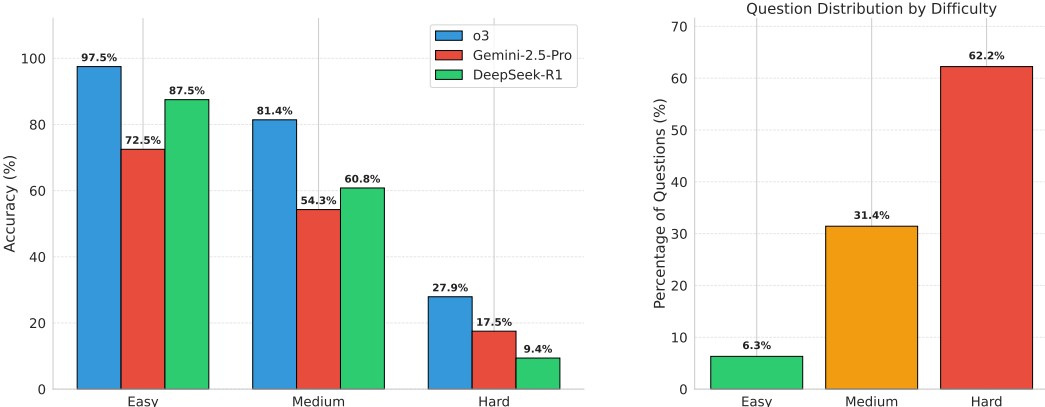

Figure 4: (a) Accuracy of the best-performing models across different difficulty levels of `Math.arXiv` and (b) distribution of difficulty levels on `Math.arXiv`. More details and results in Appendix B.2.

**Performance analysis across research subcategories.** The datasets we collect consist of mathematical problems in a wide variety of subjects. Using human-reported categories (arXiv categories, or StackExchange tags), we can get a more granular evaluation of LLMs' capabilities in specific areas. Table 4 in Appendix B.4 shows the breakdown of categories for the `Math.arXiv` dataset, with a predominance of samples from *Number Theory (math.NT)* and *Combinatorics (math.CO)*, each comprising more than 20% of the dataset size. This distribution is consistent with other recent benchmarks, e.g., MathConstruct [2] and FrontierMath [9], and stems from a combination of factors (see Figure 9 in Appendix A.4): (1) a high number of papers on the topic; (2) a higher number of theorems per paper on average; (3) a higher fraction of constructive theorems with a fixed answer.

To illustrate LLM performance on `Math.arXiv`, Figure 5 shows the accuracy per category for two of the best models on this dataset, o3 and Gemini 2.5-pro. For o3, the subcategories with the highest accuracy are *Representation Theory (math.RT), Number Theory (math.NT)* and *Analysis of PDEs (math.AP)*, each obtaining more than 60% accuracy. The subcategories with the poorest performance are *Optimization and Control (math.OC), Discrete Mathematics (cs.DM)* and *Machine Learning (cs.LG)*, with the last having only 23.8% accuracy.

However, this capability profile is not uniform across models. Gemini 2.5-pro performs best on *Machine Learning (cs.LG), Optimization and Control (math.OC)* and *Probability (math.PR)*, and worst on *Combinatorics (math.CO), Representation Theory (math.RT)*, and *Group Theory (math.GR)*. Notably, the two best categories for Gemini 2.5-pro—*Machine Learning* and *Optimization and Control*—are the ones on which o3 obtained the lowest scores!

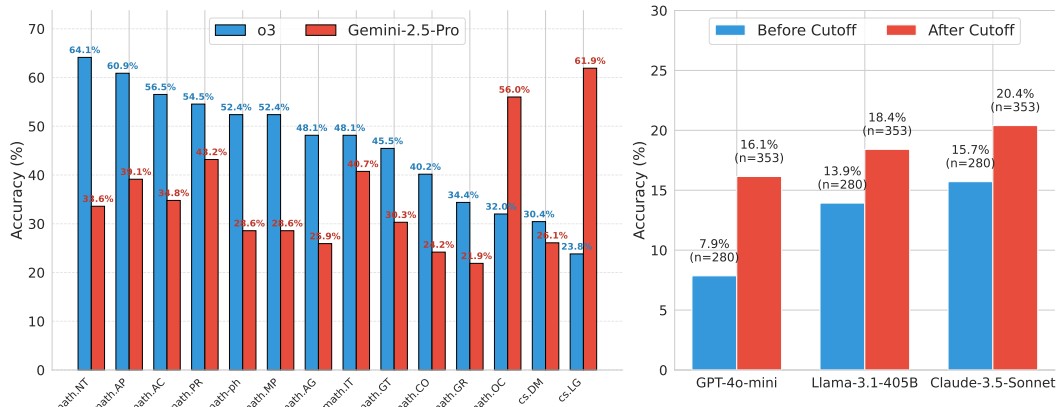

Figure 5: Evaluation of model performance on the `Math.arXiv` dataset across domain-specific and temporal dimensions. **Left**: Accuracy across mathematical domains for two models, showing significant variation by topic. **Right**: Accuracy of GPT-4o-mini, Llama-3.1-405B, and Claude-3.5-Sonnet on questions published before vs. after their training cutoff.

Overall, we observe that o3 performs well on highly theoretical tasks, such as the ones from *Representation Theory*, whereas Gemini 2.5-pro is best in domains with more practical applicability, such as in machine learning or optimization.

### 4.3 Additional Analysis

**Analyzing LLM's mistakes.** To better assess model performance, we introduce a pipeline that compares the LLMs' output with a human-authored solution, taken from the original paper or verified StackExchange answer. These are provided to the judge model o3-mini, which evaluates whether the LLMs' reasoning aligns with the human solution. Common failures include arithmetic errors, incorrect arguments, missing critical insights, or logical inconsistencies. In Figure 6 we show the error breakdown for o3, Gemini 2.5-pro and DeepSeek R1 on the `Math.arXiv` dataset. The majority of errors come from flawed reasoning, followed by conceptual misunderstanding and missed insights. This suggests that even the best-performing models struggle with multistep logical coherence in a solution.

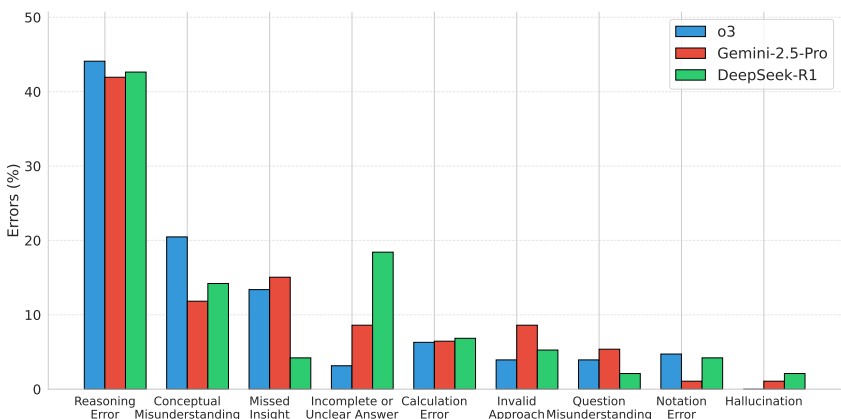

Figure 6: Error types for o3, Gemini 2.5-pro and DeepSeek R1 on the `Math.arXiv` dataset.

**How important is a theorem's context?** Recall that for questions in `Math.arXiv` and `CS.arXiv`, we provide the LLM with the full context of the theorem, namely the entire research paper preceding the theorem statement. This context may be critical to understanding the notation and statement of the theorem. However, surprisingly, we find that LLMs still perform rather well even when this context is omitted. This suggests that many theorems use concepts or notation that can be understood or inferred in isolation. For example, in the `CS.arXiv` dataset, the o4-mini model achieves an accuracy

> **Question 1:** Let $\mathcal{F}$ be an $n \times m$ Ferrers diagram with $m \geq n$, and let $1 \leq d \leq n$ be an integer. Assume that the pair $(\mathcal{F}, d)$ is MDS-constructible and let $\kappa = \kappa(\mathcal{F}, d)$. What is the value of $\lim_{q \to +\infty} \delta_q(\mathcal{F}, \kappa, d)$?
>
> ---
>
> **Question 2:** Given parameters $\sigma_1^2$, $\sigma_2^2$, and $c \leq c(\sigma_1^2, \sigma_2^2)$, what is the value of $\lim_{n \to \infty} \frac{C_s(\sigma_1^2, \sigma_2^2, c\sqrt{n}, n)}{n}$ expressed in terms of $\sigma_1^2$, $\sigma_2^2$, and $c$?
>
> ---
>
> **Question 3:** What is the value of $\lim_{\text{odd } k \to \infty} \frac{\alpha\big(f_{\{(k+1)/2\}, k}\big)}{\rho\big(f_{\{(k+1)/2\}, k}\big)}$?

Figure 7: Questions solvable by LLMs without any context. Interestingly, LLMs can infer the meaning of $C_s$ in Question 2 and of $\alpha, \rho, f$ in Question 3. Appendix B.5 contains examples of questions LLMs cannot solve without context.

of 21.6% despite the absence of any context (compared to 42.3% when the full context is provided). We present some examples of questions solvable without context in Figure 7.

**Measuring the impact of data contamination.** One of the core features of REALMATH is that it can be continuously updated with new organic samples, keeping the benchmark up to date with contemporary mathematics and preventing data contamination. This characteristic distinguishes REALMATH from other research-level benchmarks, which either require extensive manual curation by experts [9, 16], or which can only be extended by a few new samples per year [2, 19].

To test for data contamination, we split the `Math.arXiv` dataset into two batches, according to the cutoff dates of GPT-4o-mini, Llama-3.1-405B, and Claude-3.5-Sonnet. Figure 5 shows the accuracy for papers published before and after the respective cutoff dates (i.e., papers from 2022 vs. 2025). Interestingly, we find that models perform *better* on newer samples from 2025 than on older samples from 2022. One explanation could be that mathematical research became significantly easier (at least for LLMs) from 2022 to 2025, which seems unlikely. Rather, we believe a likely explanation is that current LLMs have been trained on a large amount of math data collected close to their cutoff date, and thus that LLMs may be more familiar with the type of topics studied in current math research.

We note that a nice application of REALMATH would be to study the evolution of mathematical research (and LLMs' performance) over time, an analysis which we leave for future work.

**Fine-tuning impact analysis.** To evaluate whether REALMATH can be used to improve mathematical capabilities of LLMs, we fine-tuned GPT-4o-mini on 500 random samples from the `Math.arXiv` dataset and evaluated its performance on the remaining 133 samples. Surprisingly, fine-tuning did **not** lead to an improvement in accuracy (see Appendix B.1 for more details). This suggests that the benchmark's difficulty does not stem from questions being out-of-distribution for current LLMs. Rather, current models appear to lack the specialized mathematical knowledge or skills necessary to solve certain problems, and these cannot be effectively acquired through simple fine-tuning.

## 5 Conclusion

Our work introduces REALMATH, a novel benchmark for evaluating LLMs in research-level mathematics. We show that current frontier models demonstrate surprisingly strong capabilities in research mathematics, with leading models achieving accuracy rates of 43–49% on `Math.arXiv` papers and up to 70% on `Math.StackExchange` questions. This suggests that these models may already serve as valuable assistants in mathematical research contexts, even as they continue to struggle with the most challenging problems.

A core feature of REALMATH is that it is not a *fixed* benchmark, but rather a refreshable data collection pipeline that reflects current mathematical practice while protecting against data contamination. We believe REALMATH can serve as a valuable tool for both evaluating and improving LLMs' mathematical capabilities. Its design aligns with the actual needs of mathematicians, helping advance AI assistants for research. As models improve on our benchmark, LLMs may see broader use as collaborative tools in mathematical research and education.

# 6 Acknowledgment

J.Z. is funded by the Swiss National Science Foundation (SNSF) project grant 214838. K. N. is supported by an ETH AI Center Doctoral Fellowship. We sincerely thank Yani Zhang and Meng Ding for their insightful discussions during the early stages of our data collection pipeline. We also thank Javier Rando and Daniel Paleka for their help in ideating and experimenting with different approaches for classifying problem difficulty levels. Finally, we are grateful to the anonymous reviewers for their valuable feedback and constructive suggestions.

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

# A  Data Collection and Evaluation

## A.1  Prompts for Judge Models

We used the following two system prompts fed to an LLM (o3-mini) for: (1) filtering theorems in order to retain only the ones with unique answers; (2) generating high-quality question-answer pairs from theorems.

```
1  SYSTEM_PROMPT_THEOREM = r"""
2  You are an expert in mathematics and computer science. Your task is to
       verify if a theorem has a single, numerical answer, easy to be
       verified. The theorems should be at least graduate level.
3
4  The theorems should have a fixed numerical answer, not an
       approximation. Some common examples:
5  - Necessary and Sufficient Conditions: e.g., "X holds if and only if
       condition A holds" only when at least one of A and X is specific,
       numerical quantity. We want results of the form "If condition A
       holds, then condition X holds" ONLY WHEN X is a NUMERICAL VALUE.
       We don't want "if some conditions are met, then the quantity
       satisfies a particular equation, then we can get X" when X is not
       a strict numerical value relation, because this does not have
       fixed unique solutions. Please be very strict about these rules!
6  - Existence and Uniqueness Theorems: e.g., "There exists a unique X
       that satisfies A.", but we don't want "There exists an X that
       satisfies A", because the latter is not a fixed unique solution.
7  - Exact Formula Calculations: e.g., "The answer of formula (1) is 10",
        or "The solution for formula (1) is X", then both are fixed
       unique solutions.
8  - Unique Maximum/Minimum Points: e.g., "The maximum value of function
       f is 10 at point x=1", but we don't want "The maximum value of
       function f is at least 10", because the latter is not a fixed
       unique solution.
9  - Exact Complexity Results in Computational Complexity: e.g., "The
       time complexity of algorithm A is exactly $\Theta(n^2)$" (not $\
       Omega(n^2)$ or $O(n^2)$, because big-O and big-omega are not exact
       ).
10 - Explicit number of solutions of an equation: e.g. "X has a unique
       solution y \in Y" is accepted even if the numerical value of the
       number of solutions is not specified because it can trivially be
       deduced that the number of solutions is 1, which is a fixed answer
       . We also accept "If X, there are no solutions y \in Y" (implies 0
        solutions). BUT we DON'T WANT the previous examples if the set Y
       in which we look for answers is not clear.
11 - Equality of two numerical equations: e.g., \sum_{k=1}^n k^2 = \frac{
       n(n-1)}{2} because we can assume the numerical fixed answer to be
       the difference of the 2 which is 0. You MUST include these
       equalities even if $n$ is not fixed but rather a variable. You
       MUST also include equations of the form "limit of f(n) = integral
       of g(x)"
12
13 Some examples of theorems that we don't want:
14 - We DON'T want the theorems that contain if and only if when neither
       of the sides is numerical ($x \in T$ does not represent a
       numerical value), e.g. "A graph is bipartite if and only if it
       contains no cycles of odd length."
15 - We DON'T want theorems of the type "A holds if and only if there
       exists x such that X(x) holds", but we DO WANT "A holds if and
       only if for all x, X(x) holds", where X(x) is a fixed numerical
       value.
16 - We DON'T want the theorems that have any approximations, or any
       inequalities, or any other non-deterministic statement. e.g. The
       theorems for which the main result involves the Big-O notation, or
        where the main result proven in the theorem is that a certain
```

```
        relation holds "if and only if n \geq x or n \leq y" MUST be
        rejected. We DO NOT consider any theorems where the answer is not
        an equality or a fixed answer, i.e. results of the type "n \geq 5"
         should NOT be considered, so just SKIP these types of theorems.
17  - We DON'T want the theorems that state "X $\in$ complexity class Y"
        since Y can belong to a bigger complexity class Z, so the answer
        is not unique.
18  - We DON'T want the theorems that state "X is isomorphic or
        homomorphic with Y", e.g., Chinese Remainder Theorem.
19
20
21  Important guidelines:
22  - if you cannot find a single, definitive answer, you should return an
         empty result
23  - please be very strict about the theorem, if there is any ambiguity,
        you should return a "false"
24  - Respond only in the specified JSON format
25
26  return in this exact JSON format:
27  {
28      "single_unique_answer": "true" if the theorem has a single,
            definitive answer, otherwise "false"
29      "explanation": "explanation of if this theorem has a single,
            definitive answer, otherwise an empty string",
30  }
31  """
```

```
1   SYSTEM_PROMPT_GENERATE_QA = r"""
2   You are a skilled problem setter for graduate-level mathematics and
        theoretical computer science. You are provided with a set of
        theorems (called theorems_dataset), each of which has already been
         verified to contain a single, definitive, and numerical answer.
3
4   Your task is to convert each verified theorem into a precise **
        question-answer (QA) pair**. MAKE SURE TO NOT MENTION THE ANSWER
        TO THE QUESTION IN THE QUESTION ITSELF.
5
6   Your outputs must follow these rules:
7   1. The **question** should be a well-posed mathematical or theoretical
         problem that is **clearly understandable to a graduate-level
        student**. Do not ask questions that are easy to answer without
        any mathematical reasoning or easy to guess the answer. **You must
         never begin your question with "Prove that"**
8   2. The **question must be solvable in principle with a unique
        numerical or mathematical answer**, based solely on the
        information in the theorem.
9   3. The **answer** must be:
10     - Strictly and uniquely determined.
11     - Expressed as a number, closed-form expression, formula.
12  4. DO NOT introduce extra assumptions or background. Use only what is
        stated or implied clearly by the theorem.
13  5. If a question naturally follows the structure of an identity (e.g.,
         "What is the sum of ...?"), frame it that way.
14  6. All QA pairs must reflect **the exact scope of the theorem**. Do
        not generalize, weaken, or strengthen its claim.
15  7. DO NOT generate a QA pair if the theorem is ambiguous. DO NOT
        generate a QA pair for theorems where the main result to be proven
         is an inequality or a Big-O notation. We MUST NOT include any
        kind of inequalities, questions about the lower/upper bounds, or
        any asymptotic running time of algorithms, i.e. do not generate QA
         pairs for theorems where the main result is of the type "n \geq
        5".
16  8. DO NOT include in the question the answer to it, e.g. if a theorem
        states "The limit of X is equal to Y," where Y is an expression of
```

some parameters defined earlier in the theorem, phrase the question in the manner "What is the limit of X in terms of the given parameters?", with the associated answer "The limit of X is Y". DO NOT formulate questions in the form "Prove the following relation..." since the answer will be already included in the question. Moreover, if the main result of the theorem inquires about the value of a parameter for which a relation holds, do not mention this result in the question itself but rather ask "What is the value of the parameter for which the relation holds?".

9. If you have a theorem where it says that a certain equation has a certain number of solutions (single/unique solution, no solutions, an infinity of solutions, etc.) but the acutal value of the solution is not given, consider the Question-Answer pair to be of the form "What is the number of solutions to this equation?", i.e. even if the theorem does not have an explicit numerical expression for the answer, you can consider it to be a theorem with a fixed-answer, where the fixed-answer is the number of solutions of the equation. However, if the numerical or closed-form of the solution is mentioned in the theorem statement, it is preferred to formulate the question to be "What is the solution to the following equation...?" rather than to inquire about the number of solutions.

10. For theorems of the form "X has a certain property if and only if Y has a certain property", pose the question in such a way so that the answer is the side of the "if and only if" statement which indicates a clear, numerical expression and not an abstract definition, i.e. if a theorem states "X is Pareto optimal if and only if \Phi(X) = 0", consider the question to be "If X is Pareto optimal, what is the value of \Phi(X)?" and the answer to be "\Phi(X) = 0", since the relation \Phi(X) = 0 is a numerical one, whereas the other part of the "if and only if statement", namely "X is Pareto optimal" is an abstract property. Hence, you must NOT consider the question to be "How is X if \Phi(X) = 0?" since the answer "X is Pareto optimal" is not a unique one.

11. For theorems of the form "The following identity holds: X = Y", if the identity has both X and Y as mathematical expressions that are neither closed-form, nor fixed numerical values (i.e. if we have equality between two sums with complex formulas rather than an equality of the type "X=5"), do not ask the question "What is the value of X?" and the answer to be "Y", since Y might not be the unique answer to the question. Rather, you must formulate the question to be "What is the value of X-Y?" and the fixed, clear answer to be "X-Y = 0". In this pool of theorems with this explicit QA pair, do not consider theorems where X is assumed to be a limit and Y the value of the limit, since this case should be treated as in Condition 8.

12. For theorems of the form "The following identity holds: X = Y + ct", if the identity has both X and Y as mathematical expressions that are neither closed-form, nor fixed numerical values, you should ask the question "What is the value of X - Y?" and the answer should be "ct" and not "What is the value of X - Y -ct?"

13. For theorems of the form "If X holds, then Y", formulate the QA pair to inquire about what happens with Y when relation X holds, and not under what conditions Y holds (since the condition X might not be unique if the theorem is not of the form if and only if).

14. DO NOT generate a QA pair for theorems where the main result is the belonging to a complexity class.

15. If the theorem states a result of the type "Y = |X|", where |X| indicates the cardinality of the set X, formulate the question to be "What is the cardinality of X in terms of ...?" and the answer to be "Y", but not the other way around, i.e. DO NOT state "What is the value of Y?" and the answer to be "cardinality of X", since it does not make sense from a logical point of view to phrase the question about a numerical quantity rather than about the characteristic of a set that you must determine.

```
24
25  Return your output strictly in the following JSON format:
26  {
27      "question": "Clearly stated, unique-answer question derived from
            the theorem. if the theorem is not good, return an empty
            string",
28      "answer": "The single, unique, exact answer derived from the
            theorem. if the theorem is not good, return an empty string",
29      "is_good_theorem": "true" if the theorem is good, otherwise "false
            "
30  }
31  """
```

## A.2 Prompts for Evaluation Model

We used the following system prompt and user prompt fed to GPT 4o for evaluating whether an LLM's answer to a mathematical question is correct or not.

```
1  system_prompt = r"""You are an expert mathematician tasked with
       evaluating the correctness of an answer to a mathematical question
       . Compare the generated answer to the ground truth answer and
       determine whether the generated answer is mathematically correct
       and equivalent to the ground truth.
2
3  Please be very strict and rigorous in your evaluation, mark the answer
        as incorrect even if it is 80% or 90% correct. Ensure the
       generated answer can be directly rendered in standard LaTeX
       without requiring custom command definitions. Be precise and focus
        on mathematical correctness, not formatting or style differences.
        Your evaluation should be fair and consider that the same
       mathematical content can be expressed in different ways."""
4
5  user_prompt = f"""QUESTION: {question} GROUND TRUTH ANSWER: {
       ground_truth} GENERATED ANSWER: {final_answer} Carefully evaluate
       whether the generated answer is mathematically correct and
       equivalent to the ground truth. Your response should only contain
       a JSON object with the following fields:
6  {{
7      "is_correct": boolean,
8      "explanation": "A concise explanation of why the answer is correct
            or incorrect, in a clean LaTeX format"
9  }}
10 where is_correct is true if the answer is mathematically correct and
       equivalent to the ground truth, and false if it isn't.
11  """
```

## A.3 Manual Review

For the datasets we analyze in Section 4, we rely on a human-based review to ensure the quality of the samples. As noted in Section 3.2, for `Math.arXiv` we removed 5.2% bad samples, whereas for `CS.arXiv` we discarded 7.5% samples. Figure 8 shows the percentages of manually discarded items that fall into each category (35 samples in total for `Math.arXiv` and 9 samples for `CS.arXiv`). These samples were either theorems or QA pairs that did not have a unique answer but which LLMs failed to filter out in the previous stages of the automated pipeline.

## A.4 Category Distribution Across Construction Stages

Questions in our datasets are grouped into human-labeled categories based on their domain (e.g., arXiv paper categories). Figure 9 shows how category distribution shifts throughout the dataset construction stages. Combinatorics (math.CO) becomes increasingly dominant in later stages, as

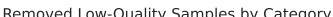

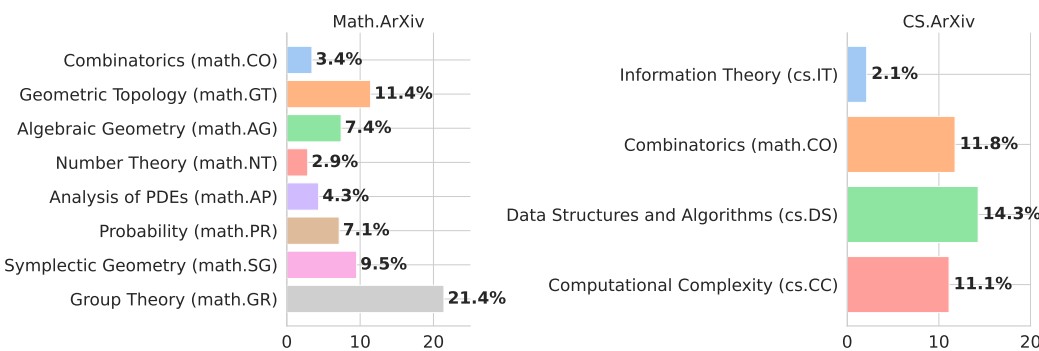

Figure 8: Samples eliminated in the final manual review phase from each category.

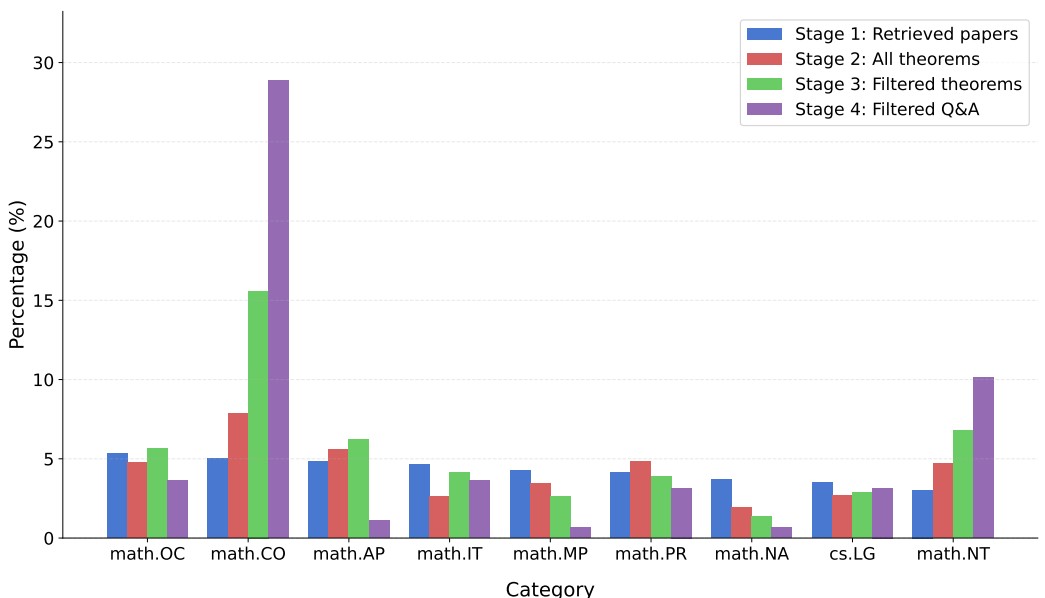

Figure 9: Category distribution across different stages of dataset construction for `Math.arXiv` (05/2022 – 09/2022). The x-axis shows subject categories (e.g., math.CO, math.NT, etc.), and the y-axis represents the percentage of the dataset each category comprises. Notably, combinatorics (math.CO) becomes increasingly dominant in later stages, particularly in the final Q&A dataset.

these papers tend to contain a large number of theorems, many of which are well-suited for the Q&A format.

### A.5  Pipeline for Mathematics Stack Exchange

The pipeline we describe in Section 3 is designed for arXiv papers. To retrieve and process samples from Mathematics Stack Exchange, we need to adapt some steps. We first queried the corresponding API to retrieve users' questions with tags such as *[limits], [definite-integrals], [integration]*—which were expected to have a clear numerical answer—as well as the top-ranked answer, if available. We extracted the HTML-formatted text and used an LLM to process the user's post into a statement and proof format, mirroring the theorems extracted from arXiv papers. We then applied a manual filtering step to ensure statements have a fixed answer and that the answer is indeed correct (for this, we checked cited sources, as well as the top-rated answer posted by other users). Finally, we fed the formulated theorem to an LLM to generate a QA pair and filter trivial samples.

### A.6 Optional LaTeX Conversion

To improve human readability of our datasets, we incorporated a LaTeX formatting step where we ask a LLM to convert each theorem into a self-contained and valid LaTeX code (this requires, for example, the expansion of any macros). We verified that this step does not negatively impact the language model's ability to formulate a QA pair or filter out low-quality samples.

# B  Additional Results

## B.1  Finetuning

In Figure 10, we use OpenAI's fine-tuning service to fine-tune GPT-4o-mini on our dataset. We present the detailed training and test loss in Figure 11. To ensure reliability, we repeat the evaluation five times and report the mean and standard deviation of the results. Surprisingly, we observe no statistically significant improvement.



Figure 10: Accuracy of original vs. fine-tuned GPT-4o-mini on `Math.arXiv`.

Figure 11 shows the training and testing loss curves for OpenAI fine-tuning. The training loss decreases rapidly and remains consistently low, indicating that the model is fitting the training data well. On the other hand, the testing loss exhibits a fluctuating behavior and increases over time compared to the beginning. This pattern may justify the failure of generalization of the model and further reinforces the idea that fine-tuning does not lead to improved performance.

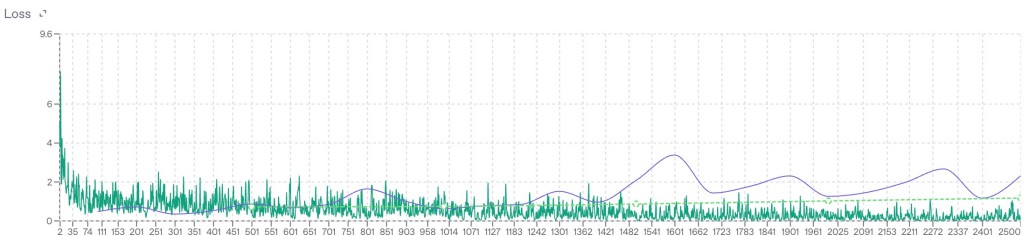

Figure 11: Training (green) and testing (purple) loss curves from the OpenAI fine-tuning service. Fine-tuning on our dataset does not show a clear improvement.

## B.2 Difficulty Levels

We assessed the difficulty level of the samples in the datasets by using older, weaker models and checking their performance on the samples. More specifically, we evaluated the performance of four older models from distinct families—Qwen3-235B, Claude 3.5 Haiku, GPT-3.5 Turbo and Llama 3.3-70B.

Samples in the *Hard* category are the ones which have not been answered correctly by any of the aforementioned models, whereas samples in the *Medium* and *Easy* categories are the ones which have been answered correctly by less than 50% of the LLMs (i.e. 1-2 out of 4), and more than 50% of the models (i.e. 3-4 out of 4), respectively.

Figure 12 depicts the distribution of samples across categories in the CS.arXiv data collection, with 69.4% of the samples being considered *Hard*. This distribution suggests that indeed our dataset reflects a hardness level appropriate for the research communities.

Moreover, we show the accuracy per category for some of the best-performing models—o3, DeepSeek-R1, Gemini-2.5-Pro—on CS.arXiv. Each of the three LLMs (except o3) has accuracy level decreasing substantially as the difficulty level increases, e.g., DeepSeek-R1 has 100% accuracy on *Easy* samples and only 11.7% accuracy on *Hard* samples. This is an expected pattern which reinforces the correct assessment of our samples in these three categories.

The higher accuracy of o3 on *Medium* samples when compared to that of *Easy* samples—75% versus 71.4%—does not necessarily imply an abnormal behavior and can be justified by the difference in the number of samples from these two categories, where we have 18% of the samples considered *Medium* difficulty and only 12.6% of them labeled as *Easy*. Especially for the *Hard* category, we spot a discrepance between the relatively high accuracy of o3, namely 31.2%, compared to DeepSeek-R1 and Gemini-2.5-Pro, who both achieve less than 12% accuracy.

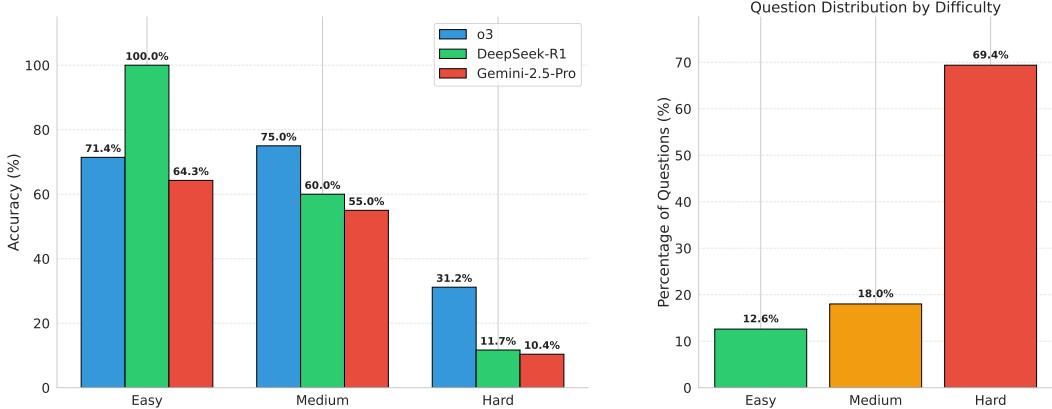

Figure 12: Accuracy across different difficulty levels for best-performing models on CS.arXiv.

Along the same line of thought, Figure 13 shows the distribution for the `Math.StackExchange` dataset. We remark that the distribution for this dataset is more spread, with the *Medium* difficulty questions being the ones that occur the most, as opposed to the `Math.arXiv` and `CS.arXiv` distributions, where the questions were dominantly classified as *Hard*. This reflects the nature of `Math.StackExchange`, which focuses on clarifications of undergraduate and graduate-level concepts and computations, rather than innovative, research-level inquiries.

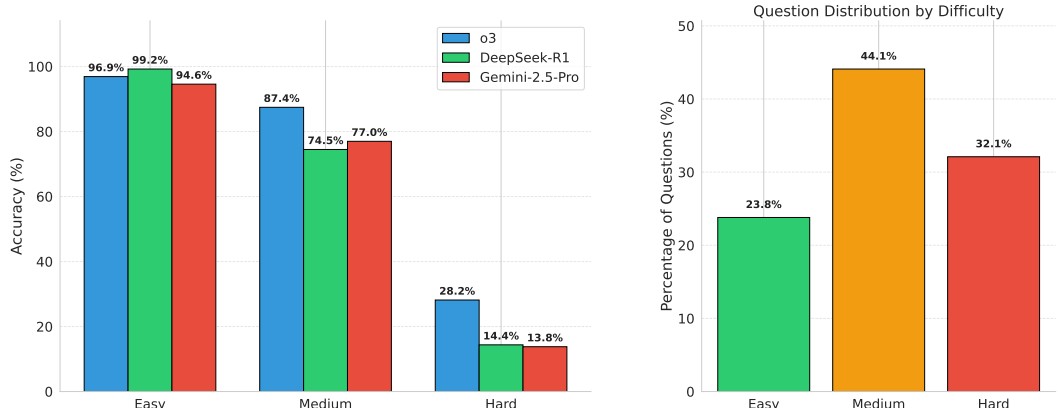

Figure 13: Accuracy across different difficulty levels for best-performing models on `math.stackExchange`.

## B.3 Performance Across Categories

We have assessed the performance of the models on each subcategory (for `CS.arXiv`) and on each tag (for `Math.StackExchange`) in order to understand whether some models perform particularly well on a certain category of questions. In Figure 14, we show the accuracies of o3 and Gemini-2.5-Pro.

On one hand, we remark for `CS.arXiv` that, with the exception of *Logic in Computer Science (cs.LO)*, where the models have identical accuracies, o3 outperforms Gemini-2.5-Pro in every category. On the other hand, an important feature that can be observed in Figure 14(a) is that o3 is better at solving questions from applied categories such as *Discrete Mathematics (cs.DM)* and *Machine Learning (cs.LG)*, while Gemini-2.5-Pro gets its highest accuracies on theoretical topics such as *Computational Complexity (cs.CC)* and *Information Theory (cs.IT)*. This is consistent with the behavior that these two LLMs have for the `Math.arXiv` dataset.

For the `Math.StackExchange` dataset in Figure 14(b) we observe as well that o3 outperforms Gemini-2.5.-Pro in each category. Moreover, the two LLMs follow a similar trend in the sense that their accuracy level per subcategory can be roughly ordered in the same hierarchy structure. Both models perform best on *Limits*, suggesting this is a relatively easier category where they can get fixed numerical answers correctly with high probability—more than 80%—, whereas their performance decreases on more advanced topics such as Complex Analysis.

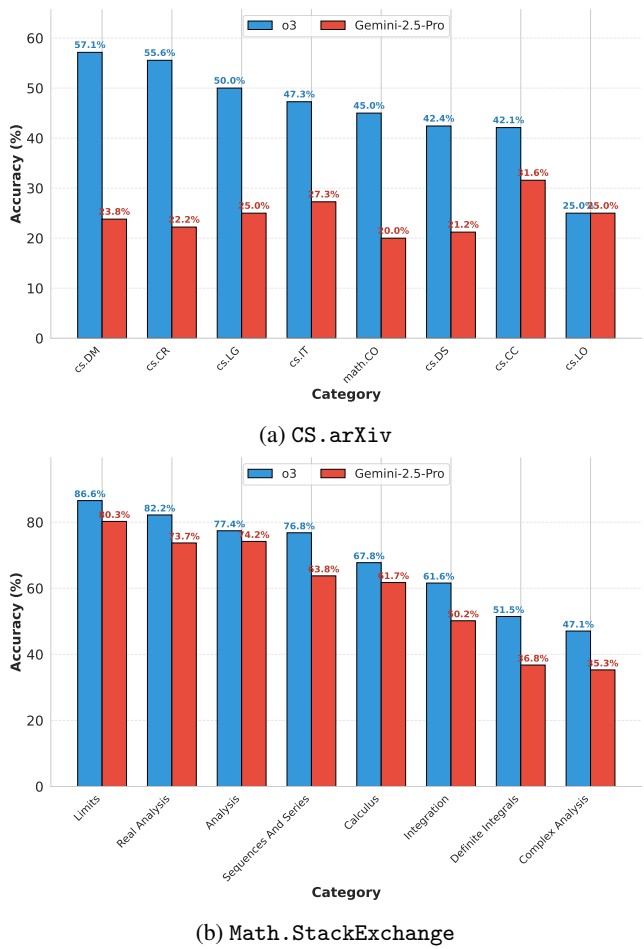

(a) `CS.arXiv`

(b) `Math.StackExchange`

Figure 14: Performance across different math domains in (a) `CS.arXiv` and (b) `Math.StackExchange`.

## B.4 Category Distribution of Questions

This subsection provides the statistics of how samples are distributed according to subdomains for each of the three datasets under consideration, clearly presented in the subsequent three tables. The subdomains are a good indicator for highlighting which mathematical domains are best for retrieving theorems along with question-answer pairs with a fixed, unique numerical answer or closed-form expression.

An interesting observation is that *Combinatorics (math.CO)* is among the top frequent domains for both `Math.arXiv` and `CS.arXiv` data collections. This suggests that problems in combinatorics are not only well-represented but also more likely to have fixed-answer solutions which satisfy our benchmarking criteria and make the automated evaluation process easy to be verified.

| Category | arXiv Tag | Percentage (%) |
|---|---|---|
| Combinatorics | math.CO | 28.89 |
| Number Theory | math.NT | 10.16 |
| Algebraic Geometry | math.AG | 5.19 |
| Probability | math.PR | 3.16 |
| Geometric Topology | math.GT | 4.51 |
| Group Theory | math.GR | 3.16 |
| Information Theory | math.IT | 7.22 |
| Optimization and Control | math.OC | 3.61 |
| Discrete Mathematics | cs.DM | 4.51 |
| Analysis of PDEs | math.AP | 1.13 |
| Commutative Algebra | math.AC | 1.35 |
| Machine Learning | cs.LG | 3.16 |
| Representation Theory | math.RT | 1.35 |
| Mathematical Physics | math.MP | 0.68 |
| Rings and Algebras | math.RA | 1.35 |
| Symplectic Geometry | math.SG | 3.61 |
| Functional Analysis | math.FA | 2.93 |
| Classical Analysis and ODEs | math.CA | 2.48 |
| Differential Geometry | math.DG | 1.58 |
| Complex Variables | math.CV | 1.58 |

Table 4: Percentage distribution of the top 20 most represented arXiv categories in the `Math.arXiv` (05/2022 – 09/2022) dataset. Categories are not mutually exclusive; papers with multiple tags are counted under each corresponding category. The predominance of questions in combinatorics and number theory is consistent with other recent math benchmarks (e.g., [2, 14]) and likely reflects an abundance of constructive statements in these topics.

| Category | arXiv Tag | Percentage (%) |
| --- | --- | --- |
| Information Theory | cs.IT | 49.55 |
| Combinatorics | math.CO | 36.04 |
| Data Structures and Algorithms | cs.DS | 29.73 |
| Discrete Mathematics | cs.DM | 18.92 |
| Computational Complexity | cs.CC | 17.12 |
| Machine Learning | cs.LG | 10.81 |
| Cryptography and Security | cs.CR | 8.11 |
| Logic in Computer Science | cs.LO | 7.21 |
| Computer Science and Game Theory | cs.GT | 3.60 |
| Optimization and Control | math.OC | 3.60 |
| Formal Languages and Automata Theory | cs.FL | 2.70 |
| Logic | math.LO | 2.70 |
| Databases | cs.DB | 1.80 |
| Algebraic Geometry | math.AG | 1.80 |
| Neural and Evolutionary Computing | cs.NE | 1.80 |
| Commutative Algebra | math.AC | 1.80 |
| Distributed, Parallel, and Cluster Computing | cs.DC | 1.80 |
| Systems and Control | cs.SY | 0.90 |
| Multiagent Systems | cs.MA | 0.90 |
| Artificial Intelligence | cs.AI | 0.90 |
| Information Retrieval | cs.IR | 0.90 |
| Networking and Internet Architecture | cs.NI | 0.90 |
| Numerical Analysis | cs.NA | 0.90 |
| Rings and Algebras | math.RA | 0.90 |
| Probability | math.PR | 0.90 |

Table 5: Percentage distribution of arXiv categories in the `CS.arXiv` dataset. Categories are not mutually exclusive; papers with multiple tags are counted under each corresponding category.

| Category | Percentage (%) |
|---|---|
| Integration | 53.32 |
| Limits | 43.91 |
| Calculus | 33.76 |
| Definite Integrals | 25.09 |
| Real Analysis | 21.77 |
| Sequences and Series | 12.73 |
| Complex Analysis | 6.27 |
| Analysis | 5.72 |
| Improper Integrals | 5.35 |
| Solution Verification | 5.35 |
| Multivariable Calculus | 4.43 |
| Derivatives | 4.43 |
| Limits Without Lhopital | 4.06 |
| Closed Form | 3.32 |
| Trigonometry | 2.95 |
| Convergence Divergence | 2.77 |
| Probability | 2.77 |
| Special Functions | 2.77 |
| Trigonometric Integrals | 2.58 |
| Contour Integration | 2.03 |
| Measure Theory | 2.03 |
| Indefinite Integrals | 2.03 |
| Complex Integration | 1.85 |
| Gamma Function | 1.85 |
| Epsilon Delta | 1.66 |
| Asymptotics | 1.66 |

Table 6: Percentage distribution of the top 26 most represented tags in the `Math.stackExchange` dataset. Categories are not mutually exclusive; questions with multiple tags are counted under each corresponding category.

## B.5 How Context Improves LLM Performance on Mathematical Problems

Figure 15 depicts some examples of questions from `CS.arXiv` dataset which are correctly solved if the relevant context is provided to the language model, but which are not answered properly when context is not provided.

Based on a manual verification of the samples, we observed that not providing context negatively impacts models in two directions:

- LLMs cannot answer the question since they do not understand the notations of certain quantities, which are only explained in the paper. In these cases, the context is crucial for the LLM since otherwise it either attempts to guess what a notation means, or they simply do not try to answer and specify that the quantities are not known to it.

- LLMs cannot answer the question because it is considered hard if we do not provide context. This situation comprises cases where the context presents some intermediate results or examples of proofs which are used for building up the current theorem or for its proof. Feeding the context to the language model improves its approach into correctly proving the question.

---

**Question 1:** For $r \geq 5$, what is the value of $\mathrm{gcover}(\mathrm{BF}(r))$ expressed in terms of $r$?

➤ *LLMs fail to correctly respond because the notation for the quantity $gcover(BF(r))$ is ambiguous without prior definition.*

---

**Question 2:** What is the expression for the zero-rate error exponent $E^{(1)}(0)$ in terms of a maximization over $q \in \mathcal{P}(\mathcal{X})$ and the function $d_B(x, x', P)$?

➤ *LLMs do not have knowledge of the function $d_B(x, x', P)$ or the set $\mathcal{P}(\mathcal{X})$ if the context is not provided.*

---

**Question 3:** Let $n > 1$ be an integer and let $\mathcal{A} \subseteq \mathbb{Z}_2^n$ be a maximal anticode of diameter one. What is the maximum possible number of codewords in $\mathcal{A}$?

➤ *The LLM knows what an anticode represents, but the question is too hard to be answered correctly.*

---

**Question 4:** Consider the complete bipartite graph $K_{m,n}$ with parameters satisfying $m < n$, $n$ even, and $(m + n)|mn$. What is the ATN of $K_{m,n}$?

➤ *The notion of ATN (Alon-Tarsi Number) is a common abreviation for language models, but the difficulty of the question is too high for them to respond correctly.*

---

Figure 15: Examples of questions in our `CS.arXiv` dataset that are not solvable by LLMs without the necessary context, but which become solvable when the context is given.

