# OpenReview forum: "RealMath: A Continuous Benchmark for Evaluating Language Models on Research-Level Mathematics"
_NeurIPS.cc/2025/Datasets_and_Benchmarks_Track — NeurIPS 2025 Datasets and Benchmarks Track poster_

### Official Review · Reviewer_MpAK · 2025-06-04

**Rating:** 4
**Confidence:** 4

**Summary:**

This work introduces RealMath, a benchmark designed to evaluate the mathematical capabilities of large language models (LLMs). Experimental results demonstrate that RealMath poses a significant challenge even to state-of-the-art LLMs. Our approach tackles three key challenges: sourcing diverse research-level mathematical content,  enabling reliable automated evaluation via verifiable statements, and  designing a continuously updated dataset to reduce contamination risks. Experiments across multiple LLMs reveal surprising proficiency in research-level mathematics compared to competition-style problems, suggesting that current models could already serve as useful tools for practicing mathematicians.

**Dataset Code Accessibility:**

Yes

**Ethical Comments:**

Yes, we can access the dataset.

**Ethical Considerations:**

No, there are no or only very minor ethics concerns

**Final Justification:**

There is no concerns.

**Limitations Weaknesses:**

* The QA pair number is limited, which is less than 1500
* The dataset name is RealMath. However, it seems that the dataset still needs manual annotation.
* The question and answer context lengths are short. Therefore, the dataset cannot be used to evaluate math problems that require long context.
* **It is not clear whether the LLM training data already contains the RealMath data.**

**Strengths Contributions:**

* This work introduces a new framework for assessing LLMs' mathematical capabilities using *organic research-level mathematics*, moving beyond traditional competition-style problems.
* This work proposes a contamination-resistant methodology for constructing and maintaining mathematical benchmarks, ensuring long-term relevance.
* The analysis reveals how current LLMs perform on tasks mirroring real-world mathematical research, highlighting their potential and limitations.
* The dataset presents a substantial challenge—state-of-the-art models achieve **<30% accuracy** in experiments, underscoring the gap in advanced reasoning.
* Structured with **graded difficulty levels** (easy to hard), the dataset enables granular analysis of LLMs' mathematical proficiency across skill tiers.
* The dataset spans **multiple mathematical categories**, with detailed distributions, facilitating targeted evaluation of model capabilities.

---

> ### Author Rebuttal · Authors · 2025-07-31
>
> We thank the reviewer for their valuable time and comments. We appreciate the overall positive feedback and would like to briefly address the questions and weaknesses raised as follows.
>
> 1. The QA pair number is limited, which is less than 1500
>
> Thanks for noting this. Although the dataset currently contains fewer than 1,500 QA pairs, it prioritizes quality and diversity. Our pipeline also supports continuous updates to expand the dataset over time. The current number of QA pairs reflects the output from running the arXiv extractor over a one-year period. By extending the extraction window to cover more time, we can easily increase the number of QA samples in the dataset.
>
> 2. The dataset name is RealMath. However, it seems that the dataset still needs manual annotation.
>
> Our pipeline itself does not require human evaluation. However, we did manually review the 633 theorems, as well as the generated QA pairs, to verify consistency, mathematical correctness, and the validity of having a unique correct answer. Manual annotation is a nice-to-have, presenting a trade-off between improved data quality and increased annotation effort
>
>
> 3. The question and answer context lengths are short. Therefore, the dataset cannot be used to evaluate math problems that require long context.
>
> Could you provide more details or context regarding this point?
>
> To clarify, before presenting the question to the model, we provide all the preceding text of the theorem as context. This means the LLM has access to the full relevant context needed to answer the question, even if the question itself is short. We believe this approach enables the model to effectively handle long context by incorporating all necessary information prior to the question.
>
> 4. It is not clear whether the LLM training data already contains the RealMath data.
>
> Thanks for the comment. While we cannot completely exclude overlap with LLM training data, REALMATH is designed to continuously add new problems published after the models’ training cutoffs, which helps reduce contamination. Our pipeline allows evaluation on the latest data to further minimize overlap.

---

> > ### Comment · Reviewer_MpAK · 2025-08-04
> >
> > Thank you for the response. I decide to keep the score.

---

### Official Review · Reviewer_vokq · 2025-07-03

**Rating:** 4
**Confidence:** 4

**Summary:**

This paper introduces RealMath, a benchmark consisting of 633 instances designed to evaluate LLMs on research-level mathematical problems. It focuses on constructive problems sourced from arXiv papers and Math StackExchange, where solutions can be easily verified based on the final answer. The authors propose a pipeline for data acquisition, filtering, and rewriting mechanisms, enabling the dataset to be continuously updated. Experiments on the benchmark show that LLMs achieve accuracy rates of 43–49% on problems from Math.ArXiv, and up to 70% on Math.StackExchange questions.

**Dataset Code Accessibility:**

Yes

**Dataset Code Comments:**

The dataset, along with instructions for running LLMs on it, is publicly available on GitHub.

**Ethical Considerations:**

No, there are no or only very minor ethics concerns

**Final Justification:**

Given that RealMath is specifically limited to constructive problems and its evaluation may involve potential data contamination, I nonetheless acknowledge its potential as a continually evolving pipeline that could be extended to more problems. After reviewing the other reviewers' comments, I have decided to maintain my original score.

**Limitations Weaknesses:**

- Although the paper claims to target research-level mathematics, it focuses exclusively on constructive problems—a relatively narrow and less challenging subset of the domain.
- There is a significant risk of data contamination, as the benchmark sources problems from online platforms. The context-based experiments seem to support this concern.
- The conclusion and analysis regarding contamination based on test samples from 2022 vs. 2025 may be unreliable. The observed higher accuracy on 2025 samples could simply result from those problems being easier, rather than indicating lower contamination.

**Strengths Contributions:**

- The paper is novel, well-motivated, and well-written—it was a pleasure to read.
- The proposed pipeline supports continuous data collection, enabling the benchmark to be regularly updated and remain reflective of evolving mathematical practice.

---

> ### Author Rebuttal · Authors · 2025-07-31
>
> We thank the reviewer for their time and comments. We now make some brief comments to address the questions and weaknesses raised above.
>
> 1. Although the paper claims to target research-level mathematics, it focuses exclusively on constructive problems—a relatively narrow and less challenging subset of the domain.
>
> We acknowledge that our current benchmark primarily focuses on constructive problems, which constitute a more concrete and well-defined subset of research-level mathematics. This focus was intentional, as it allows for clear problem formulations and reliable automated evaluation—both critical for scalable and consistent benchmarking. That said, our pipeline is not limited to theorems with explicit fixed-answer formats. Rather, we include any theorems that can be transformed into a form with a unique, verifiable answer.
>
> We agree that expanding RealMath to incorporate more open-ended and abstract problems is a valuable and important direction for future work. We plan to explore methodologies to include such problem types while maintaining rigorous and meaningful evaluation standards.
>
> 2. There is a significant risk of data contamination, as the benchmark sources problems from online platforms. The context-based experiments seem to support this concern.
>
> We acknowledge the concern about potential data contamination. To mitigate this risk, our pipeline is designed to flexibly extract up-to-date evaluation data, it's not a fixed benchmark.
>
> For example, if a new model is released in June 2025, we can collect fresh math papers from arXiv in July 2025 for evaluation, thereby minimizing overlap with training data. This dynamic evaluation capability allows us to reduce the risk of contamination over time.
>
> 3. The conclusion and analysis regarding contamination based on test samples from 2022 vs. 2025 may be unreliable. The observed higher accuracy on 2025 samples could simply result from those problems being easier, rather than indicating lower contamination.
>
> Thank you for this insightful comment. We agree that the observed higher accuracy on 2025 samples could potentially be influenced by differences in problem difficulty, and we cannot fully exclude this possibility at this stage. However, we consider it unlikely that the mathematical problems from 2025 are systematically easier than those from 2022, given the steady growth and complexity of research in the field.
>
> We acknowledge that fully disentangling data contamination from inherent difficulty differences requires further analysis, which we leave for future work.

---

> > ### Comment · Reviewer_vokq · 2025-08-05
> >
> > Thank you for the response. Given that RealMath is intentionally and exclusively focused on constructive mathematics problems, I recommend revising the title, abstract, and introduction to more accurately reflect this specific scope, rather than presenting the work as broadly addressing general mathematical research.
> >
> > While I acknowledge that some degree of data contamination may be unavoidable in the current dataset and evaluation, I agree that the authors' efforts could still be valuable for future research in this area.
> >
> > Overall, given the current state of the paper, dataset, and evaluation, I decide to maintain my original rating.

---

### Official Review · Reviewer_1vLR · 2025-07-04

**Rating:** 4
**Confidence:** 4

**Summary:**

This paper introduces REALMATH, a novel benchmark designed to evaluate LLMs on authentic research-level mathematics sourced directly from arXiv papers and Math Stack Exchange. Addressing limitations of existing competition-based benchmarks, REALMATH features automatically curated question-answer pairs with verifiable answers (e.g., closed-form expressions) extracted from theorems. Its key innovation is a continuously refreshable pipeline that mitigates test contamination by incorporating new research. Experiments reveal that while frontier LLMs (e.g., o3) achieve moderate accuracy (e.g., 49.1% on Math.arXiv), performance drops sharply on hard problems (27.9%), highlighting both potential utility as research assistants and significant remaining gaps in formal reasoning.

**Dataset Code Accessibility:**

Yes

**Ethical Considerations:**

No, there are no or only very minor ethics concerns

**Limitations Weaknesses:**

1. The theorem-to-QA conversion process (Figure 3) seems similar to concepts presented in the recent paper[1], and I look forward to the authors adding a discussion of paper[1] to the camera ready version.
2. I am still concerned about the data collection process that is fully automatic and does not involve human annotation, and I look forward to seeing more experiments or discussions about data quality.

[1] One Example Shown, Many Concepts Known! Counterexample-Driven Conceptual Reasoning in Mathematical LLMs.

**Strengths Contributions:**

1. Introduces REALMATH, the first benchmark derived directly from research papers (arXiv) and math forums (Math Stack Exchange). It features automatically curated, verifiable QA pairs from theorems, capturing authentic research-level math beyond competition-style problems.
2. Proposes a continuously refreshable pipeline that automatically sources new problems from arXiv/StackExchange. This enables regular updates with post-training-date content, mitigating test-set contamination risks inherent in static benchmarks.
3. Benchmarks show frontier LLMs (e.g., o3: 49.1% on Math.arXiv) surpass performance on competition benchmarks (e.g., MATH, IMO). This suggests LLMs may already aid real mathematical research, despite struggling with the hardest problems (27.9% accuracy on hard subset).

---

> ### Author Rebuttal · Authors · 2025-07-31
>
> We thank the reviewer for their time and comments. We now make some brief comments to address the questions and weaknesses raised above.
>
> 1. The theorem-to-QA conversion process (Figure 3) seems similar to concepts presented in the recent paper[1], and I look forward to the authors adding a discussion of paper[1] to the camera-ready version.
>
> Thank you for pointing this out. The dataset in [1] primarily consists of textbook problems extracted using OCR tools, with annotation performed by annotators holding bachelor's degrees in applied mathematics. In contrast, our RealMath dataset is sourced from arXiv and StackOverflow, and does not require manual annotation. We agree that a comparison is valuable and will include a more detailed discussion of [1] in the related work section of the camera-ready version.
>
> 2. I am still concerned about the data collection process that is fully automatic and does not involve human annotation, and I look forward to seeing more experiments or discussions about data quality.
>
> While our pipeline doesn’t rely on human evaluation, we manually reviewed all 633 theorems and the corresponding QA pairs to ensure consistency, mathematical soundness, and the uniqueness of correct answers. We found that approximately 6% did not meet our quality criteria and were filtered out.
> We think that human evaluation is a valuable addition, though it comes with a trade-off between data quality and annotation cost.

---

> > ### Comment · Reviewer_1vLR · 2025-08-05
> > **Official Comment by Reviewer 1vLR**
> >
> > Thank you for the response. I decide to keep my score.

---

### Official Review · Reviewer_6zuc · 2025-07-05

**Rating:** 5
**Confidence:** 4

**Summary:**

This paper introduces REALMATH, a novel benchmark for evaluating LLMs on research-level mathematical tasks. The benchmark collects data from internet platforms such as arXiv, addressing challenges in acquiring authentic content, enabling automated verification, and preventing data contamination. It conducts comprehensive testing on various state-of-the-art LLMs.

**Additional Feedback:**

- Does REALMATH systematically cover all mathematical subdomains? Is there a systematic taxonomy in place?
- Given that arXiv papers may contain errors, how does REALMATH verify the correctness of theorems?

**Dataset Code Accessibility:**

Yes

**Ethical Considerations:**

No, there are no or only very minor ethics concerns

**Limitations Weaknesses:**

### Weaknesses
- The approach relies on automated pipelines and LLMs for data filtering, lacking in-depth verification of mathematical correctness.
- Dependence on LLMs (e.g., o3-mini) for theorem filtering and QA generation introduces potential biases and randomness due to model limitations.

### Limitations
- ArXiv papers, being non-peer-reviewed, may contain errors, and the benchmark assumes correctness without verification.
- Using LLMs for answer evaluation introduces subjectivity and randomness.
- The benchmark is limited to verifiable QA pairs, excluding existential proofs, inequalities, and other problem types.

**Strengths Contributions:**

- REALMATH extracts problems from real research papers and forums, reflecting mathematical research practices more authentically than competition-focused benchmarks.
- The automated data construction pipeline ensures scalability and mitigates potential data contamination risks by dynamically acquiring the latest data.

---

> ### Author Rebuttal · Authors · 2025-07-31
>
> We thank the reviewer for the time and comments. We appreciate the positive feedback and would like to make some brief comments to address the questions and weaknesses raised above.
>
> 1. The approach relies on automated pipelines and LLMs for data filtering, lacking in-depth verification of mathematical correctness.
>
> Thank you for this question. As discussed in our paper, while our data collection pipeline is primarily automatic, we did include a human evaluation step in the end. We reviewed manually the 633 theorems, as well as the QA pairs produced (to check consistency, mathematical correctness, validity of the unique answer to the question, etc).  Human evaluation is a nice-to-have, presenting a trade-off between improved data quality and increased annotation effort.
>
> 2. ArXiv papers, being non-peer-reviewed, may contain errors, and the benchmark assumes correctness without verification. Given that arXiv papers may contain errors, how does REALMATH verify the correctness of theorems?
>
> While our data construction process assumes the correctness of mathematical theorems sourced from arXiv papers, we do acknowledge that this assumption is not guaranteed.
>
> We could consider the occasional ambiguities in statements or notation to be a feature rather than a flaw, as they reflect the authentic, in-the-wild nature of real mathematical practice—unlike the more polished and curated content typically found in formal competition or textbook settings.
>
> And restricting our pipeline to only peer-reviewed sources could help ensure higher accuracy, but it would come at the cost of reducing the diversity, breadth, and timeliness of the material we can include.
>
> 3. Does REALMATH systematically cover all mathematical subdomains? Is there a systematic taxonomy in place?
>
> Yes, REALMATH systematically covers a wide range of mathematical subdomains. We retrieved all arXiv papers with "math" as the primary category, and included the top 20 most represented arXiv subcategories in the Math.arXiv dataset (see Table 4 in the appendix). These subcategories follow arXiv's subject taxonomy, which serves as our classification framework.

---

> > ### Comment · Reviewer_6zuc · 2025-08-09
> > **Thanks**
> >
> > Thanks for your rebuttal, I am still postive to this paper.

---

> ### Comment · Area_Chair_6PY4 · 2025-08-06
>
> Please take a look at the author's rebuttal and respond accordingly.
>
> Thanks
>
> AC

---

### Decision · Program_Chairs · 2025-09-18

**Decision:**

Accept (poster)

**Comment:**

This paper introduces REALMATH, a novel benchmark for evaluating LLMs on research-level mathematics using an automated, continuously refreshable pipeline with authentic problems from arXiv and Math Stack Exchange.
The reviews are mixed but leaning positive (one Accept, three Borderline Accepts), with consensus on the benchmark’s novelty and community value; earlier concerns about data verification, scope, and contamination have been convincingly addressed.
Given its strong novelty, resolved concerns, and clear community impact, the AC recommends Accept.